# Microstructure and Anisotropic Order Parameter of Boron-Doped Nanocrystalline Diamond Films

Somnath Bhattacharyya 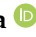

Nano-Scale Transport Physics Laboratory, School of Physics, and DST/NRF Centre of Excellence in Strong Materials, University of the Witwatersrand, Johannesburg, 1 Jan Smuts Avenue, Private Bag 3, WITS 2050, Johannesburg, South Africa; somnath.bhattacharyya@wits.ac.za; Tel.: +27-11-7176811

**Abstract:** Unconventional superconductivity in heavily boron-doped nanocrystalline diamond films (HBDDF) produced a significant amount of interest. However, the exact pairing mechanism has not been understood due to a lack of understanding of crystal symmetry, which is broken at the grain boundaries. The superconducting order parameter ($\Delta$) of HBDDF is believed to be anisotropic since boron atoms form a complex structure with carbon and introduce spin-orbit coupling to the diamond system. From ultra-high resolution transmission electron microscopy, the internal symmetry of the grain boundary structure of HBDDF is revealed, which can explain these films' unconventional superconducting transport features. Here, we show the signature of the anisotropic $\Delta$ in HBDDF by breaking the structural symmetry in a layered microstructure, enabling a Rashba-type spin-orbit coupling. The superlattice-like structure in diamond describes a modulation that explains strong insulator peak features observed in temperature-dependent resistance, a transition of the magnetic field-dependent resistance, and their oscillatory, as well as angle-dependent, features. Overall, the interface states of the diamond films can be explained by the well-known Shockley model describing the layers connected by vortex-like structures, hence forming a topologically protected system.

**Keywords:** boron-doped diamond; superconductivity; grain boundaries; transmission electron microscopy; electrical transport; magnetoresistance; topological insulator

## 1. Introduction

In granular superconducting systems, the fine-grain boundaries separated by grains can act as Josephson junctions. The microstructure of a granular superconducting system can be vital, unlike a single crystal superconductor, in controlling many of the transport properties, such as critical temperature, normal state resistance, coherence length, and the $H_{C2}$-T phase boundary. The crystal structure of a single and microcrystalline diamond is well-known, unlike its nanocrystalline form which contains a high density of atomically thick grain boundary regions. The effect of heavy boron doping on the microstructure of the heavily boron-doped nanocrystalline diamond films (HBDDF) has created significant interest; however, the origin of the superconducting phase, as well as its relationship to the microstructure, has not been clearly understood [1,2]. From microscopic and spectroscopic studies, the morphology of HBDDF shows a layered structure that may suggest topological phases in HBDDF films instead of conventional granular superconductors [3,4]. So far, topological superconductors are made based on heavy fermions by creating layered structures; however, such a complicated multi-element hybrid system finds difficulties in device applications today. Therefore, a need for using a single system such as diamond (or other carbon structures) is revealed by doping light elements (such as boron) in a gas phase. The symmetry of the lattice at the grain boundaries (GB) can be broken intrinsically without using a heavy magnetic material so that all superconducting transition features can be easily understood in a simple hybrid system. Since the microstructure of the diamond films is associated with non-sp$^3$ carbon (particularly, at the GB) [5] and other defects, hence unconventional superconductivity and a magnetic phase in HBDDF can be observed [6].

In this report, the microstructure and electrical properties of HBDDF are significantly different from conventional granular superconductors. Such interesting superconducting behavior related to strong spin-orbit coupling in BNCD films has not been found in any other carbon systems such as graphene (since the spin-orbit coupling is weak). Regarding unconventional superconductivity (not mixed with a magnetic impurity), the Cooper pairs conducting through this atomically thin GB would be subjected to a nonequivalent electric field (defined by the interface potential as a step function), and thus a spin-orbit coupling of the Rashba-type (RSOC) followed by spin-triplet superconductivity arises [7]. However, details of the microstructure of the interface and their interconnectivity remain unclear until today. Therefore, the proposed interface RSOC needs to apply to the diamond lattice to achieve the topological phase [8,9]. In diamond GB, accommodating boron atoms or holes can produce charging effects in the layers, creating Andreev bound states (ABS). Besides, we have claimed a bound state formation in HBDDF associated with the charge-Kondo effect in the two-dimensional layered structures within diamonds [10]. Moreover, we have shown spin-triplet superconductivity in the diamond interface due to RSOC created by breaking the translational symmetry [11], where additional symmetry-breaking operations can take place [12]. RSOC at the GB produces spin channels, hence an Andreev reflection (AR) at the spin active (SA) interface can undergo a $\pi$ phase change or a sign inversion of d-wave superconductors [13,14]. Unlike a d-wave, a *p*-wave superconductor is much less studied, although intrinsic or spontaneous $\pi$-junction is such a system that has been suggested [15]. An oscillatory (exchange-type) interaction associated with the (magnet-like) defect centers can introduce a strong orbital effect, suggesting a well-defined GB aligned in the form of a lattice or a superlattice (SL) structure. These features will be tested through microstructure analysis (in Section 3.1) and electrical transport at low temperatures and magnetic fields (in Sections 3.2–3.4).

Further, we show how a theoretical model can explain the connection between the transport features and the microstructure of BNCD films, which can lead to the understanding of a topological insulator in a carbon system. The model can be extended to a generalized SL structure such as in the Fu–Kane–Mele and the Shockley model that consist of multilayers in the presence of bound states (such as ABS) coupled to each other [16,17]. The topological phase needs a modulation of the order parameter in a hybrid structure arising from the atomic potential that can be found in the *p*-wave GB structures coupled to an s-wave superconducting diamond crystal. The superposition of ABS having multifaceted properties can work as a superposition of reflected and transmitted Cooper pairs. They can be connected to create a non-local state, hence an odd-frequency $\Delta$ can be realized in a diamond hybrid SL structure as predicted earlier [18,19]. Quantum simulations of the proposed model are presented (Sections 3.4.1–3.4.3).

## 2. Materials and Methods

The HBDDF samples were grown using microwave plasma-enhanced chemical vapor deposition. This method produces dense plasma and dissociates carbon precursors (e.g., methane) to produce high-quality (metal contamination-free) diamonds (films) (see Appendix A and Figure A1). These are all grain-size dependent and the grain size can be controlled through synthesis parameters. Hence, it offers a great deal of control over the physical properties of the system. Fused quartz substrates were pre-cleaned and seeded by diamond nanoparticles before deposition using 95% of $CH_4$ (CAS number 74-82-8) in $H_2$ (1333-74-0) with 4000 ppm trimethylborane (TMB) (593-90-8) to $CH_4$. The substrate temperature was 850 °C and the pressure was ~80 Torr. The microwave power applied was 1.4 kW. The sample was prepared from 1%$CH_4$ in $H_2$ and achieved a boron concentration of $2.8 \times 10^{21}$ cm$^{-3}$ (i.e., well above the Mott metallic transition ~$3 \times 10^{20}$ cm$^{-3}$) and was composed of large grains ~50–70 nm in size. A JEOL JEBL-7001FLV field emission high resolution (1.2 nm resolution at 30 kV) scanning electron microscope (SEM) has been used for microstructure analysis (Figure 1a). The columnar growth of diamond films of ~100 nm thick is shown in Figure 1b,c.

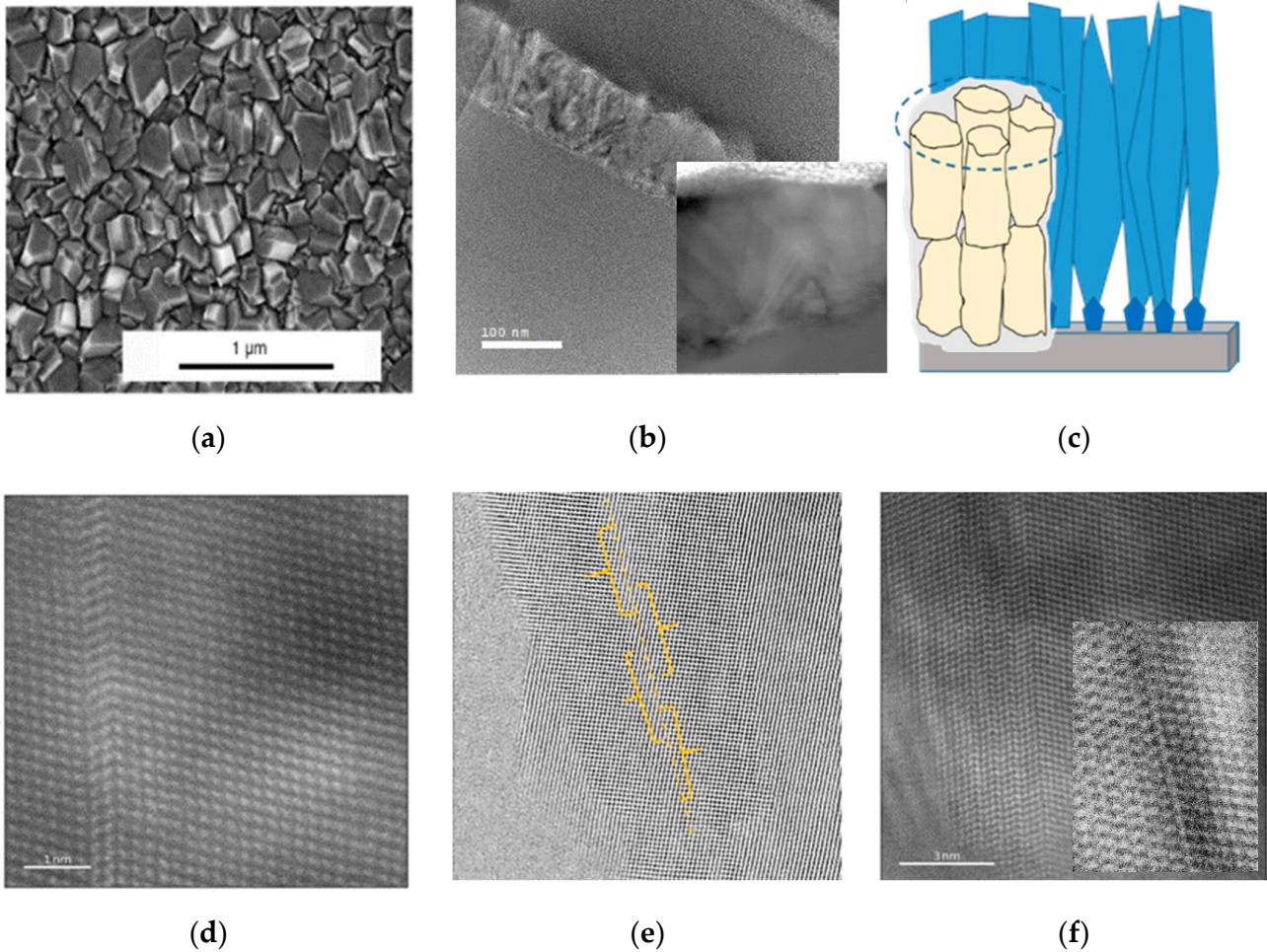

**Figure 1.** (**a**) Scanning electron micrograph shows diamond grains distribution. (**b**) A cross-section transmission electron microscopic image shows the columnar microstructure of the films. (Scale bar 100 nm) (**c**) Columnar growth of the diamond nanocrystals from the seeded substrates is shown schematically. High-resolution TEM imaging indicates high crystalline order of individual grains with sharp distinctive grain boundary edges (**d**) distortions (scale bar 1 nm) and (**e**) the modulation in width. (**f**) A close observation shows distorted hexagonal rings on the interface (scale bar 3 nm) (see inset of the figure).

Electrical resistance (*R*) and magnetoresistance (MR) measurements were performed using a cryogen-free (Cryogenic Ltd., London, UK) measurement system using a completely automated measurement system consisting of a Keithley current source meter, a nanovoltmeter, and a lock-in amplifier. The range of temperatures (*T*) was from 0.3 K to 5 K, whereas the magnetic (*B*) fields changed from 0 to 5 Tesla. Using a four-probe van der Pauw geometry, the longitudinal resistance $R_{XX}$ was measured by applying current on one edge and measuring voltage drop on the opposite end. Transverse resistance ($R_{XY}$) was similarly measured by applying a current across two diagonally opposite corners of a 5 mm × 5 mm chip and measuring the voltage across the other two diagonally opposite vertices. The *B* field was applied perpendicular to the diamond film.

The scanning transmission electron microscopy (STEM) was performed at the Diamond light source (UK) at various magnifications up to 25 million times (Figure 1d–f). For the microstructure study, the TEM lamella samples were prepared at Carl Zeiss Microscopy using an ion beam milling technique.

## 3. Results

### 3.1. Microstructure

The ultra-high-resolution TEM (UHRTEM) investigations of the grain boundaries show sharp crystal twinning and layered stacking faults (the interfacial region between grains) (Figures 1d–f and 2). Microstructure analysis of CVD-grown diamond films oriented close to the [110] zone axis (see captions for details). The grain boundaries (and the twinning) suggest the presence of bi-layer phases in the HBDDF, which finds a remarkable similarity with previous work [4]. In addition, lattice distortions due to boron doping are noticed in Figure 1d–f. Atoms are shifted from the regular position, which can form a dimer or a periodic modulation of the bond length, similar to a single and a double bond. From very close observation of the twinning of the GB, distorted five- and six-fold rings (Figure 1d) are identified. These distorted chain-like structures and the triangular shapes of the microstructures are proposed as sufficient for the simulation of the topological phase in HBDDF (see later part of the text). The microstructure looks similar to bi- or tri-layer graphene, maintaining ABA stacking where $\pi$ orbitals work as electric fields normal to the planes and bind the layers (Figure 2a–c). The superconductivity of the heavily boron-doped diamond may be compared to inhomogeneous superconductivity in disordered organic materials [20], such as vortices in the layered superconductors as a bi-layer charge column, as described in the generalized Shockley model [17]. It is well known that chemical vapor deposition (CVD) diamond films frequently show twinning where the crystal symmetries of the twins strongly affect the properties of the grain boundaries (Figure 1d–f) [21]. The effect of high boron concentrations on the lattice symmetries of the diamond is also an interesting aspect of this system. It has been observed that boron has a preferred growth direction in the diamond Figure 1b [22] and also that the boron acceptor subsystem can lead to inversion symmetry breaking [23], i.e., the formation of points of non-centrosymmetry, that have been linked to spontaneous time-reversal symmetry breaking [24] as well as a static Jahn–Teller effect [25]. There are also reports of the formation of a bilayer of boron in HBDDF that has been investigated as a possible precursor to interfacial superconductivity in this system [4,26]. These intrinsic symmetry-breaking features of the boron acceptor, as well as a crystal lattice, are known to have huge implications for the superconducting $\Delta$ of other type-2 superconductors, particularly in triplet *p*-waves [27]. In Figure 2a–c, we see the most abundant twinning is that of $\sum = 3$-grain boundaries and that higher-order boundaries occur through when such symmetries meet [21,28]. These higher-order twins ($\sum = 9$, 27, etc.) have a stronger lattice mismatch and can lead to dislocations within the boundary region (Figure 2b,d,f). It has also been established that the diamond grain's surface termination (strong lattice miss-match) can result in the formation of an extended $\pi^*$ orbital configuration due to the hybridization of dangling bonds [29]. The high tightened stress at the grain boundary region can lead to surface states through a modification of the electronic energy forming frontier electronic orbitals [30], in addition to this, there are many reports on the grain boundary conduction of various diamond systems [31,32]. All of these studies indicate that the electronic transport of granular diamond systems is greatly dependent on the grain boundary regions, this becomes particularly interesting when considering the superconducting HBDDF as granularity and boundary scattering events in superconducting systems can lead to interesting phenomena (Figure 2e,f, Figures 3–5). This is well known for type-two superconducting systems such as the high $T_c$ cuprates where transport properties of the GB have been thoroughly investigated [33–35]. Such systems can exhibit zero-bias resonances in their tunneling spectra due to the formation of bound states at the grain boundaries [34,35]. These bound states are the result of the anisotropic nature of the pair potential (*p*-wave) and the significant change that occurs as the pairs scatter from the grain boundary (Figures 4b and 5d). Some of the microstructures were shown in another report; [11] however, here we show how the microstructure is closely related to angle-dependent magnetoresistance.

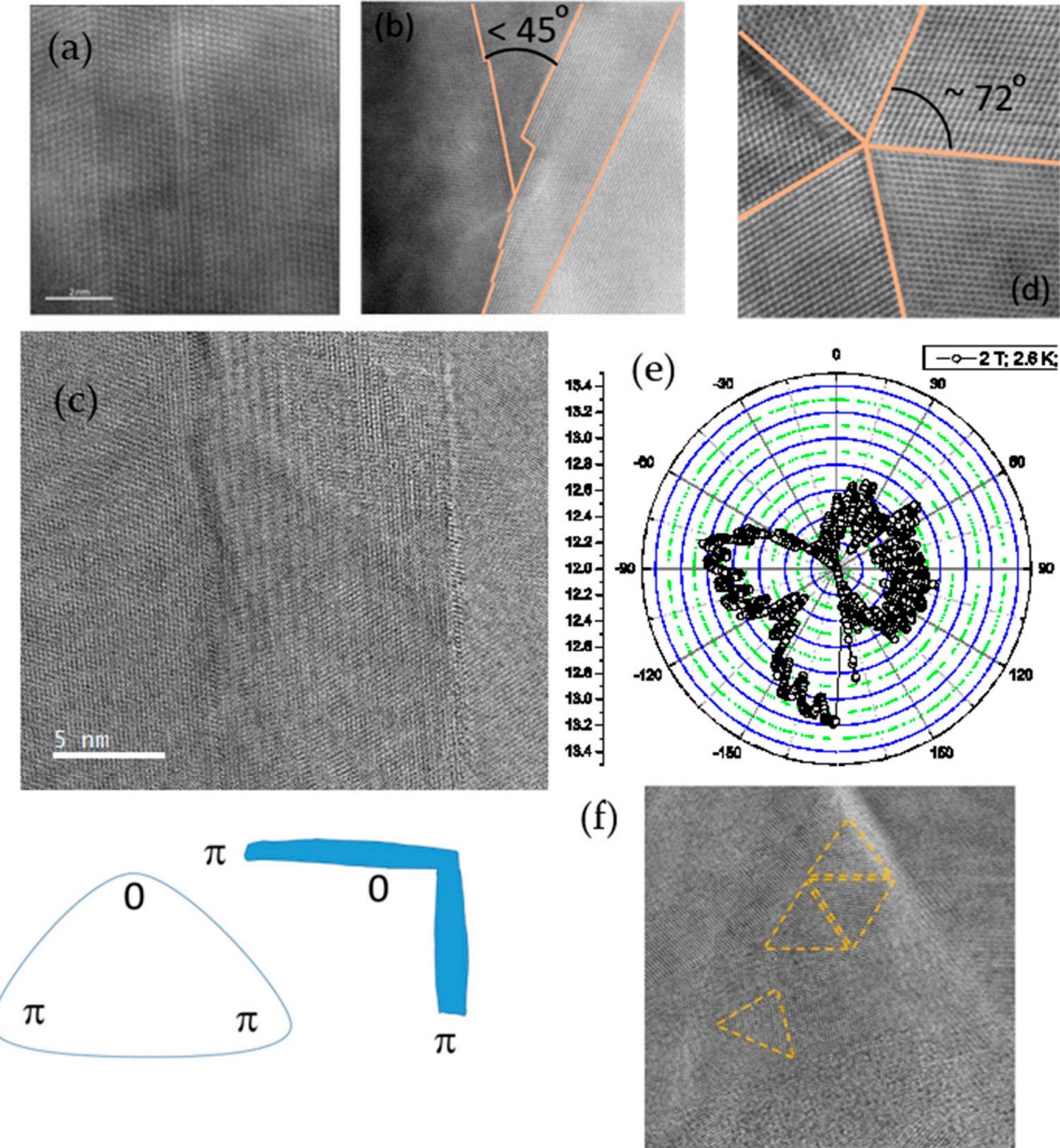

**Figure 2.** (**a**) UHR−TEM imaging of the HBDDF. The individual grains are highly ordered with clearly defined lattice planes. Grain boundaries are regular planes that span the length of the individual grains and define planes of crystal twinning. (**b**) The intersection of the grain boundaries occurs at 45° (when two twinned planes meet). (**c**) High-angle annular dark−field (HAADF) imaging allows for a more detailed structural analysis; the extended lattice is then dominated by stacking faults terminated by crystal twins. (**d**) It also shows higher-order intersections of fivefold symmetry oriented at approximately 72° apart and stacking fault layers where translational symmetry is broken. (**e**) Angle−dependent resistance (Ohm) in polar co-ordinate shows pronounced anisotropic features, with 72° as well as smaller periods, indicating grain boundary conduction. The critical temperature of the system is shifted to lower temperatures upon rotation in a small field. (**f**) A schematic indicating the effect lattice translational invariance and grain boundary mismatch can have on an anisotropic superconducting order parameter, as observed in the high $T_c$ cuprates, flipping of the pair potential can occur, this gives rise to bound states at the junction area. A triangular shape can introduce a $\pi$ phase of the Cooper pairs. However, this can happen from the (111) plane (see discussion in the text). Lines are drawn to guide the eyes.

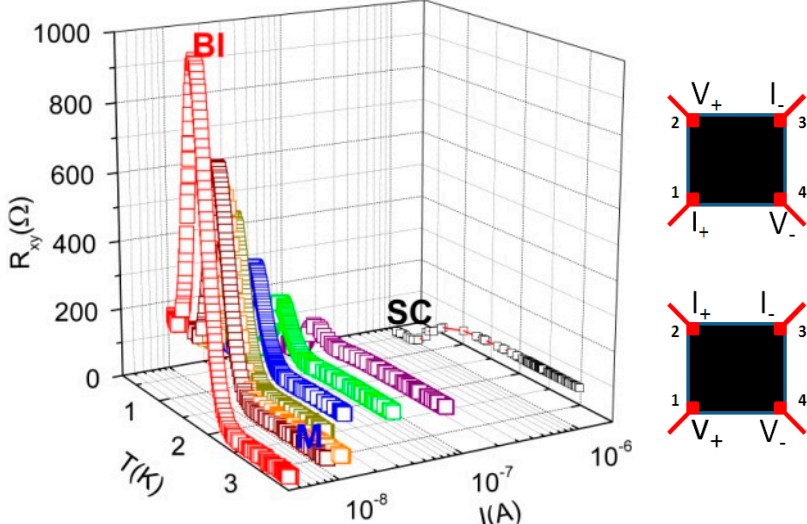

**Figure 3.** Evolution of metal-insulator-superconductor (SC) transition with a current variation. from 1 µA down to 10 nA. The resistance shows various phases as current and temperature are varied. It is seen that as the bias current is reduced the bosonic insulator (BI) peak increases sharply but does not shift in position. The geometry for the transport property measurements is shown on the right.

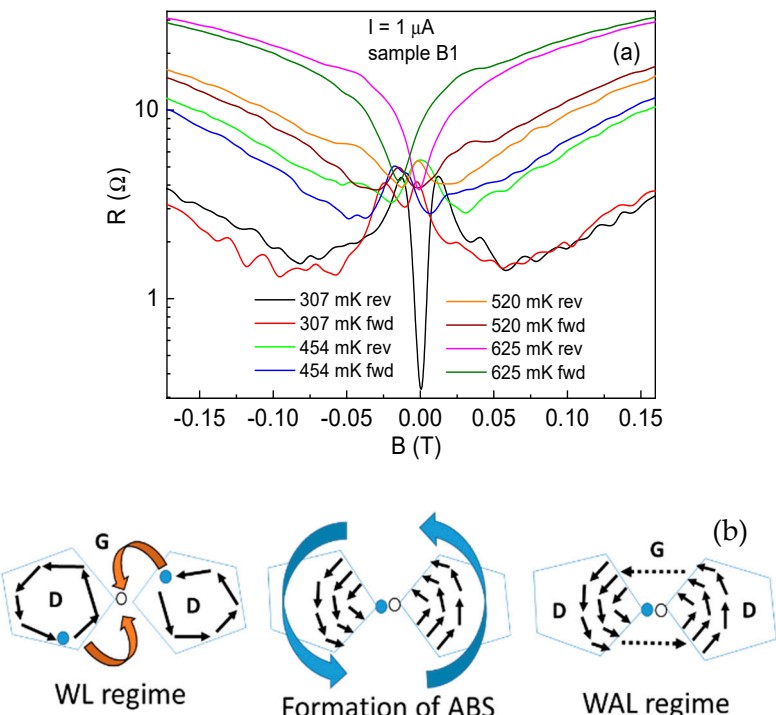

**Figure 4.** (**a**). Magnetoresistance of HBDDF changes the slope from a positive to a negative trend (at ~600 mK) as the sample temperature rises. (**b**) The left figure describes two WL paths of spin strictures (filled circles and small arrows) and the orbit−like ABS is formed by connecting the WL paths around a fixed point (open circles) that is not accessed by the spins. Effectively, this configuration can be formed by twisting a ring into another ring (large arrows). The middle figure represents an RSO configuration where the spin and orbits are mutually perpendicular to each other, which can also be described as two rings making interconnecting loops. This configuration can be described by a vortex structure with ABS at the core, as shown in the figure on the right. Here, D and G represent diamond grain and grain boundaries, respectively.

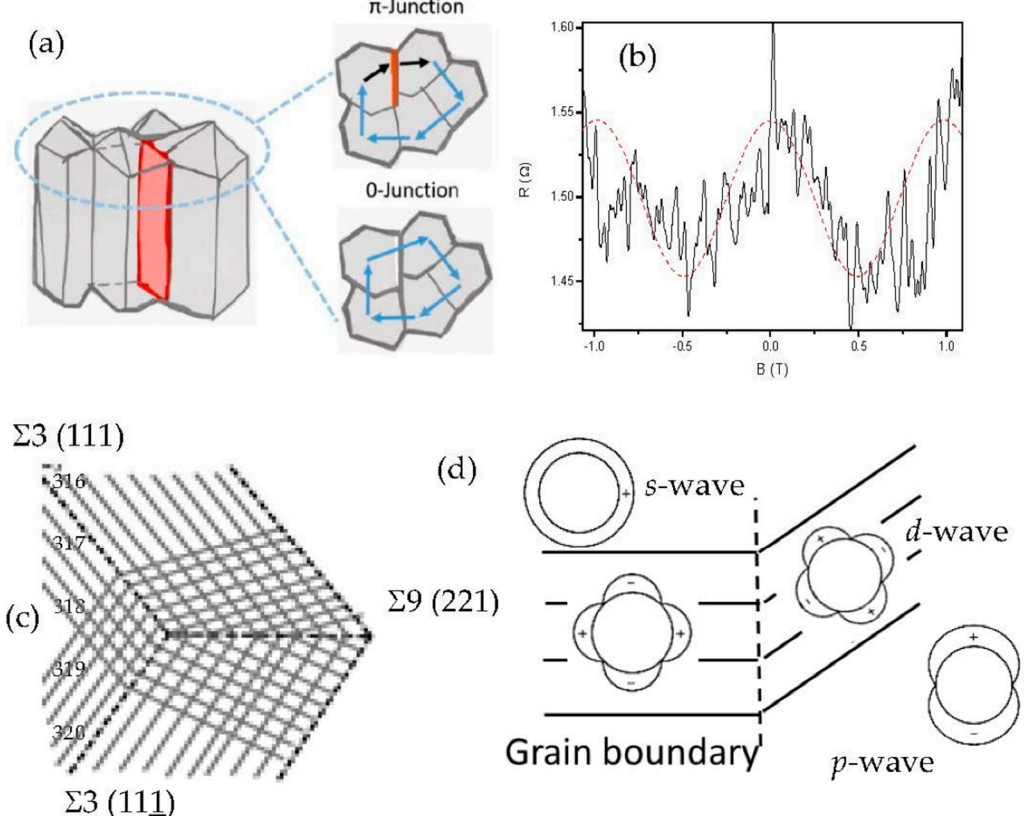

**Figure 5.** Magnetoresistance (**a**) A schematic indicating the geometry of conduction loops considered using the fitting model, which takes into account interference between 0−junction (of high transmission) and π−junctions (where phase scattering/spin flipping occurs). (**b**) An oscillatory magnetoresistance (recorded at 2.2 K) shows a more pronounced π−junction character. (**c**) Transport features are explained through the anisotropy of the crystal structures. The high proportion of ∑3 boundaries allows for higher-order twinning such as the ∑9 and this is schematically shown in (**d**) [21]. (**d**) The pairing symmetry *s*, *d*, and *p*−wave are shown as a shift of the charge distribution. (*d*−wave) Cooper pairs transmit through the grain boundaries.

Now, we shall interpret the origin of RSOC in the diamond GB structure. The symmetry of a pair wavefunction is given by momentum × spin × frequency. A key parameter to describe the symmetry breaking of the grains producing the grain boundary region is defined as $\Sigma = [C1 \cdot (C2 \times C3)]/[a \cdot (b \times c)]$, which finds similarity with the symmetry breaking of the angular momentum vector in introducing the helicity popularly known as the Rashba SOI, described by Edelstein as interface SO: $H_{SO} = \alpha(p \times c) \cdot \sigma \delta(c \cdot r)$ where c is one of the two nonequivalent normal unit vectors and the δ function describes the interface potential with a position vector (r). We see such kinds of structures in Figure 2b. As seen in Figure 2c, the periodic potential variation in the GB from the mismatch of the lattice planes gives rise to modulation of Δ. To probe the complex symmetry of HBDDF films, detailed studies of angle (θ)-dependent resistance are conducted (see Figure 2e, this includes temperature-dependent resistance at different applied fields and isotherms of MR (followed by a Shockley model simulation)). The maxima of misalignment take place at the angle of 45° (or 72°), which gives the peak in the angle-dependent MR (Figure 2b,d,e). Such phenomena cannot occur in *s*-wave superconductors due to the isotropic pair potential and, thus, their observation has been used to classify superconducting systems [36]. The present observation of a superlattice-like sub-system (Figure 2c) has appeared to be different from the boron interfacial material claimed by previous authors; however, it shows other geometrical phases such as a triangle (Figure 2f). Such microstructures have been reported

earlier (see Appendix A). These microstructures introduce a $\pi$-phase to the propagating Cooper pairs.

### 3.2. Electrical Conductivity: Evidence for RSOC

The RSOC effect in these samples is confirmed by the bias current-dependent R-T behavior that shows a pronounced peak before reaching the superconducting onset as the sample temperature is lowered (Figure 3). The intensity of the so-called boson insulator (BI) or a charge Kondo peak decreases exponentially with the bias current or the density of charge (electrons) supplied to the system [1,37]. Since the lattice symmetry in this diamond system is broken in the grain boundary region a level splitting occurs yielding RSOC. However, the increase in current populates the spilled levels and widens the band, which starts overlapping the level, and superconductivity is regained as the peak is depleted. By decreasing the supply of the carriers to the system the effect of level splitting and the RSOC becomes prominent. With the increase in the rate of scattering from the spin–orbit interactions the resistance of the samples also increases. To further investigate the nature of this BI peak the MR is measured in the same temperature range.

### 3.3. MR Transition below and above the Critical Point

As shown in Figure 4a, a transition from a positive to negative MR is observed as the temperature is increased. This shows a crossover from weak localization (WL) (corresponding to a triplet state) to weak anti-localization (WAL) (singlet state) above 500 mK [38]. As we know that the different MR regimes (WL or WAL) correspond to the scattering processes, a higher resistive state (WL) occurs due to coherent backscattering, whereas the WAL state corresponds to enhanced conduction. Like the insulator peak, this transition results from a bound state formed at the grain boundaries of the diamond and produces RSOC in the superconducting diamond films (Figure 4b) (see Figure caption for details). This also supports the modulated $\Delta$, as suggested in ABS lattices based on the triangles (as observed from the microstructure), such as a Kagome lattice (Figure 2f).

The MR of the sample is measured at different temperatures between 300 mK and 600 mK (Figure 4a) with the field applied normal to the film. We observe hysteresis at all temperatures. Qualitatively, this means the resistance increases sharply with the increasing field up to a maximum (sharp peak) and then decreases again as the field is further increased until it reaches the background curve. The hysteresis behavior has been observed in the R vs. B at different temperatures below or around the $T_c$, which is a new observation in these structures where magnetic impurities have not been introduced, therefore explaining the effect of RSOC in this system. The anomalous peak MRs of two kinds were observed, for example, one with a smooth transition and the other with asymmetric resonant peaks in the transition region (asymmetric with respect to the scan direction) (Figure 4a). This can be attributed to setting the combination of bias current and temperature to some point of instability regarding switching between superconducting and insulating states. These features can be attributed to the properties of $\pi$-Josephson junctions. The ABS is known to exist in the ab-plane of layered superconductors and at superconductor–normal metal interfaces, which explains the $\pi$-junction as shown in Figure 4b.

### 3.4. Angle-Dependent Transition

Figure 2e shows *R* vs. *T* while rotated at different angles in the presence of a 2 Tesla field, the critical temperature of the sample is observed to change as a function of the angle of the applied field. Such behavior is frequently seen in superconducting spin valves where a long-ranged triplet-state is induced due to spin mixing at magnetic interfaces [39]. As shown in Figure 2e, the magnetoresistance is highly anisotropic with a periodicity of $90°$. The angle-dependent change in the magnetoresistance can thus be qualitatively explained if we consider superconducting pairs with anisotropic pairing potential scattering off spin-polarized boundaries. The rotation of the sample in the magnetic field changes the orientation of the spin-polarized boundary with respect to the anisotropic Cooper pair, this

orientation will result in either backscattering (WL) or transmission (WAL) of the Cooper pair. In Figure 4a, the proposed *p*-wave scattering a shows sign change of $\Delta$ accounting for the transition, which needs a simulation and theoretical modeling (given later). Similar results have before been observed in layered superconductors such as La(O,F)BiSeS and are directly related to an anisotropic order parameter [40].

The nature of AR for an M/S junction is like a resonant state that makes a sign change of the pair potential on the Fermi surface, i.e., makes a $\pi$-phase shift and therefore was described as a mid-gap Andreev resonant state (MARS) (Figure 4b) [18]. This ABS or MARS can be created or activated below the superconducting onset, which exhibits a transition from the WAL effect depending on the temperature dependence of the spin-orbit scattering (explained in a previous paper, [11]). Below 500 mK, the MR becomes positive at the low B range, which shows a clear signature of RSOC. As the bias current becomes very low (50 nA), the SOC effect becomes very prominent as seen from the sharp feature at B $\rightarrow$ 0. Start with the application of the B field that can break the symmetry and allow an interplay of singlet and odd parity by breaking the singlet or, triplet and even parity by fusing or joining two odds to make an even parity state [18,19].

In Figure 5a, we concentrate on low field regions and record some oscillatory and negative MR features. If we accept the layered structure in the present HBDDF (Figures 1 and 2), then a number of observed features can at least be qualitatively explained by a 0–$\pi$ JJ hypothesis. This model includes a resonant level and is explained to be due to S-I-S structures [41]. The key concept is based on explaining observations such as negative MR in terms of negative Josephson coupling as given in [42]. The transmission of the (*d*-wave) Cooper pairs through the grain boundaries is shown in Figure 5d as the structural anisotropy is strongly related to the parity of Cooper pairs. The mid-gap bound states can only manifest in anisotropic pairing such as the *d*-wave and *p*-wave states.

### 3.4.1. Topological Superconductor

From the unusual electronic transport properties, we reveal that they can be correlated with the microstructure of HBDDF and also different from single-crystal diamond. Hence, we propose a model to explain the topological features that are developed based on the Fu–Kane model and the Schockley model of a topological insulator.

According to the Fu–Kane model, the deformation of a diamond lattice through the incorporation of boron atoms in the (111) direction may lead to a topological insulator state of a diamond lattice [16,17]. It is exactly the (111) direction reconstruction of a diamond lattice by boron doping that was claimed by Polyakov et al. [4]. Based on those observations we assume that the type of mentioned interface might be the s-wave/topological insulator state. If it is a reasonable hypothesis then, as it was shown theoretically, such types of interfaces can contribute to the overall superconductivity but with a *p*-wave order parameter due to a specific interface scattering. In other words, the boron-doped diamond should demonstrate *p*-wave superconductor phenomena in addition to the common s-wave superconductor observations as shown in Figure 5d.

We explain the topological transformations by assuming that the topology of the initial pure diamond Fermi surface is strongly affected by the boron doping process and that finally leads us to a modified Fermi surface, corresponding to the 3D topological insulator state. The problem is to establish a particular type of boron-induced deformation and, subsequently, make a connection with the corresponding topological phases. However, on a pure diamond surface, a dimerized atomic structure of the diamond (111) direction was proposed as follows [42].

### 3.4.2. Geometric Phase Acquired by the GB

From the microstructure and electrical transport, we reveal a superlattice-like structure that is probably created from the displacement of the (111) plane in the diamond induced by boron incorporation. At present, a theoretical model explaining all these microstructural features and electrical properties is not available. Due to the reconstruction of the

diamond surface described as a Pandey chain, the (111) plane is formed, which has some special properties [43,44]. This has not only a very flat band (close to graphene layers) but also contains a large density of carriers (electrons) resulting in strong electron–electron interactions, hence superconductivity and magnetism can be possible in this particular (111) layer, which leads to spin-triplet superconductivity as claimed here. The (111) plane forms a triangle since one bond is absent from the three nearest neighbors on a trigonal (planar) structure (Figure 2f) instead of a regular tetragonal structure of a diamond crystal. The atoms on the (111) surface form a 2 × 1 superstructure and a zigzag chain after the (111) plane is displaced by the boron [43,44]. The Zak phase from the band structure or dimer state (including two hopping parameters and two order parameters) can explain the origin of the topological features in superconducting diamond films. Boron atoms further introduce a topological phase to the diamond structure with more deformation along the (111) planes yielding an extremely flat energy band and an inversion symmetry (Figure 1d,e) [45,46]. We think that the UHRTEM images in Figures 1 and 2 of the interface represent the Zak phase.

### 3.4.3. Shockley Model Simulation

Based on the previous model, we propose a structure that can explain the MR due to spin-orbit coupling. The carbon atoms are arranged as a set of qubits on a quantum simulator (Figure 6a). In diamonds, boron atoms form a spin-orbit state verified by Raman spectroscopy [23–25] (Figure 6b). In addition, the formation of a new acceptor level of approximately 37 meV associated with the B–C sheets was suggested and was able to mix with the spin-orbit split diamond valence band [4]. This can be shown in a four-level energy diagram in Figure 6b. At the temperature where the effect of RSOC is not dominant due to a short spin scattering length, the effect of WL dominates where the forward and the backward scattering paths are overlapping without breaking the symmetry in twin grain boundaries, as observed from the UHRTEM microstructures (Figure 2f). The rings can be triangular like a Kagome lattice, which can also match the microstructure of the films; however, the connectivity of the edges of the triangles is important to define the vortices (Figure 6a). This model consists of two complementary planes connected by a line that represents a vortex. This is formed by alternating the bond lengths like a polyacetylene chain, however, applied to a 3D lattice, like a diamond. When the triangles are placed in a line and connected by tunnel barriers the structure can be described by an Aharonov–Bohm ring or a vortex since the two WL orbits are overlapping and produce a magnetoresistance oscillation. Due to the spin-orbit coupling, the rings can be separated and form a tube-like structure. In that case, the edges of the triangles can be connected to a tube, however, remain almost flat like a bilayer. In Figure 6c, the return probability ($P_r$) of the electrons with a detuning time can be compared to the MR with the B field described in [47–49]. The variation of $P_r$ shows an initial decrease at low fields up to a certain B field followed by a side peak. This is a signature of weak antilocalization and oscillations due to the RSOC, as obtained in Figures 4a and 5b. We have performed a holonomic operation of the qubit-rotation to capture a geometric phase and simulate RSOC, which claims topological features [48,49]. Details of the quantum simulation are given elsewhere [48,49].

Grain boundaries being atomically thick cannot be imaged properly even when UHRTEM has been employed. It is difficult to find the exact nature of the bonding arrangements at the interface that can be correlated to the topological features of transport. Nevertheless, this is the first report showing the application of superconducting diamond films in quantum devices where the junction properties can be controlled by the application of a field. We believe that the potential application of carbon-based topological qubits. We think that in carbon systems only diamond has these potential applications.

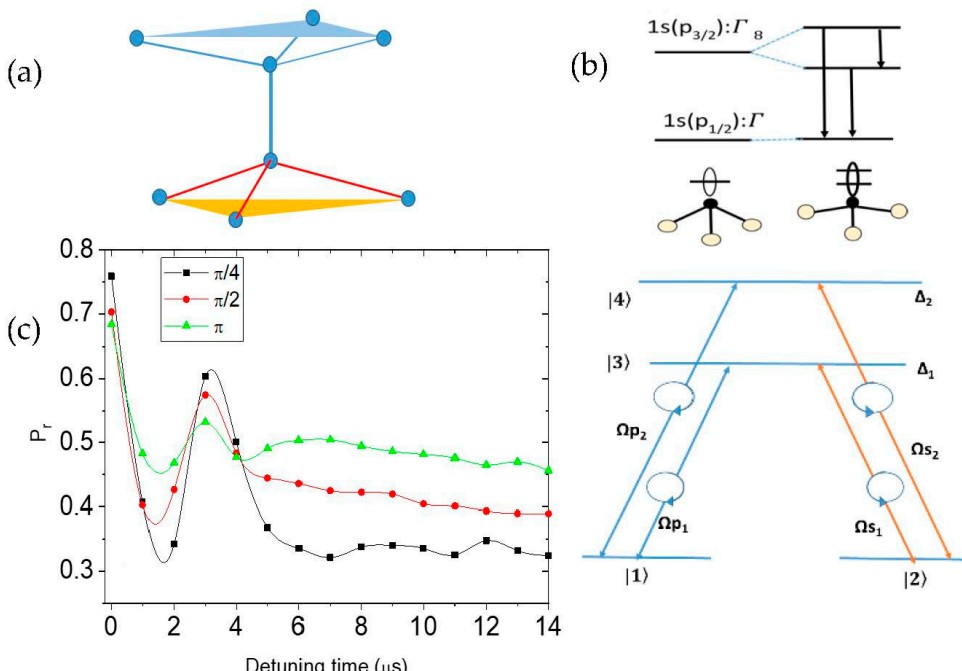

**Figure 6.** (**a**) Shockley model of topological phase [17]. Qubit arrangements show two connected triangles as closed loops and describe spin-orbit interactions. (**b**) The boron acceptor can be present in the GB, which removes the degeneracy of the $1s(p_{3/2})$ spin-orbit split level, hence creating a strong splitting of the energy levels [23]. A four-level energy ($|1>-|4>$) diagram of the spin triplets of a boron center showing the Rabi frequencies ($\Omega_{s,p}$) and the detuning is $\Delta_1 = \Delta_2 = \Delta$ [48]. (**c**) Shows the variation of $P_r$ (or magnetoresistance) with detuning time (magnetic field) when spin-orbit is included. The qubits are also rotated in different angles from $\pi/4$ to $\pi$ to show the intensity variation of the side peak, which is a signature of RSOC.

## 4. Discussion

We have searched for boron–carbon phases and a layer of boron (or borophene) in heavily boron-doped diamond films [50–52]. While we do not see any direct evidence of such a phase a very interesting structure of the grain boundary of diamond has emerged. Our investigations into the diamond microstructure revealed that the nanocrystalline diamond film morphology is fundamentally different from other granular superconducting systems. This is because, in the case of a superconducting diamond, the boron interfacial material forms a sub-system that alternates with the diamond system in a somewhat ordered fashion similar to a superlattice, while in conventional granular superconducting material there is usually no additional subsystem separating the individual superconducting grains.

In our previous work, we showed details of metallic transport in the grain boundaries of nitrogen-doped nanocrystalline diamond films [31]. The formation of this so-called anomalous bosonic insulator was ascribed to be due to confinement effects associated with the granular nature of the system. In this system, the insulating state is a result of a Coulomb blockade or repulsion between superconducting regions analogous to granular material.

However, the transport properties of boron-doped diamond films appear to be more interesting and, therefore, we attempted to match with GB features by employing some of the best microscopic techniques. The UHRTEM study reveals a superlattice-like structure of the GB. The twinning of the GB also shows a distorted filamentary structure that is produced from the distortion of the diamond lattice by boron incorporation. In addition, the locally formed triangular or loop-like structures observed from the microscopy suggest a topological phase in the materials. Quantum simulation of the structure in Figure 6a finds a remarkable similarity with the magnetoresistance features of weak anti-localization in Figure 6c of such systems.

In this work, the analysis of these data firmly establishes a non-*s* wave order parameter in these systems as well as spin-triplet superconductivity. The most important finding in this article is the long-range triplet Josephson current in these granular media, which can be demonstrated by changing both magnetic fields, applied bias current, temperature, and the applied magnetic field angle. We explored these outstanding features using a topological phase associated with a complex anisotropic order parameter. The structure of the BNCD film is similar to a *p*-wave superconductor where the sign change of the order parameter takes place between the layers. A plausible explanation of the triplet state is given based on a spin-polarized or chiral Andreev doublet state. We believe that, based on this study, a diamond-based topological qubit can be realized.

**Funding:** CSIR-NLC: the URC Wits, and National Research Foundation (SA) for the BRICS project.

**Institutional Review Board Statement:** Not applicable.

**Informed Consent Statement:** Not applicable.

**Data Availability Statement:** Not applicable.

**Acknowledgments:** SB is very thankful to Christopher Allen (Diamond Light Source, UK) for the HRTEM studies. Miloš Nesládek is acknowledged for supplying the BNCD samples. I am thankful to D. Matuko for the production of valuable transport data. C. Coleman and D. Churochkin are acknowledged for their valuable contribution to this work including some analysis and stimulating discussions. Shaman Bhattacharyya is acknowledged for the quantum simulation. Discussion with S.N. Polyakov, V. N. Denisov, and V.D. Blank was very useful for the paper. I would like to thank A. Kirkland (Oxford University) for granting HRTEM beamline time and D. Wei and C. Hyunh (Carl Zeiss Microscopy) for lamella foil preparation.

**Conflicts of Interest:** The authors declare no conflict of interest.

## Appendix A

**Synthesis of diamond films by microwave CVD:** Amorphous diamond-like films (instead of crystalline films) can be deposited by the ion-beam technique. Nanocrystalline diamond (powder instead of films) can be produced by a detonation technique. Single-crystal diamond can be produced by a high-temperature, high-pressure technique; however, doping diamond with light elements, particularly with boron, is not easy. This technique is very expensive as well. The synthesis of high crystalline diamond films requires high energy (temperature) that can be provided by a hot filament CVD technique; however, it can introduce contamination in the films from the metal electrodes. Using a plasma CVD technique, high-density plasma and a very high temperature of electrons can be produced, which can produce diamonds of a high crystalline order. Most importantly, the gas phase doping of diamond can be made in a microwave CVD system with chemicals such as TMB that introduces boron into diamond films (Figure A1). Microwave CVD is a clean technique that does not introduce any metal impurities to diamonds.

**Comparison with other studies:** Earlier HRTEM studies have revealed that the boron aggregates in granular samples, particularly in samples with a micrometer grain size, form triangular pockets between diamond crystal grains [3]. We have seen such structures in our samples.

Other studies claimed the intergranular boron network was responsible for the superconductivity instead of the diamond; however, there was no proof since the measurement represented some very localized structures [4]. Nevertheless, the formation of C–B bilayer sheets along the {111} planes was suggested. This included high order, superlattice reflections in X-ray diffraction and Laue patterns that unambiguously showed an incommensurately modulated structure due to the displacement of boron atoms.

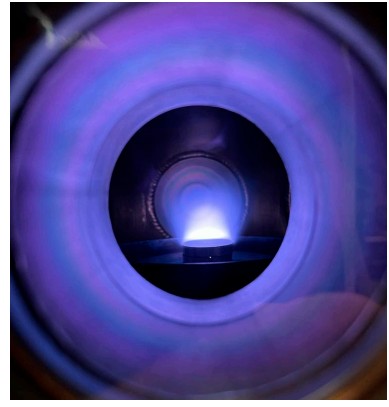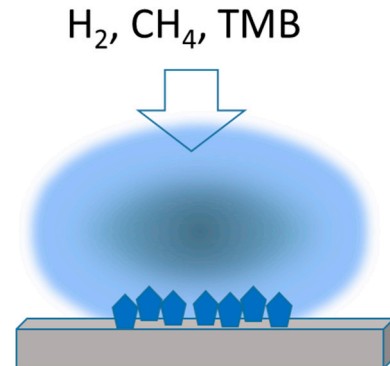

**Figure A1.** Plasma CVD deposition of diamond films (**left**) from the seeded substrates is shown schematically (**right**).

This model is based on the substitution of two carbon atoms in the (1/2, 1/2, 0) and (1/4, 3/4, 3/4) positions of the diamond unit cell by two boron atoms. Since the B–C bonds (1.6 Å) are longer than the C–C bonds (1.54 Å), the boron atoms are shifted towards each other along the [111] direction and their displacement provides the elastic strain relaxation. The boron pair changes the cubic ABCABC stacking sequence on the hexagonal CACA stacking sequence. In Pandey's symmetric chain model, the bond length of the (111) plane was similar to graphite, yielding the surface energy band, whereas two surface bands were derived from the bonding and antibonding combinations of dangling orbitals along the chain producing the bulk bandgap. The calculated Fermi surface was found to be flat and degenerate and nearly degenerate along with J-K directions with the presence of electron-hole packets and unstable against Jahn–Teller distortions. Overall, this chain is remarkably similar to polyacetylene with alternating single and double bonds like a dimer, which yields a gap at the Fermi surface due to this asymmetry [42]. Such a directionally flat (completely dispersionless) band can produce a straight-line-shaped Fermi surface as observed in type-III Dirac cones and tested from molecular-orbital representation [45]. Starting from a square-lattice model in 2D for spinful fermions, this model has also been extended to the diamond-lattice model where a type-III Weyl semimetal has been constructed. Although a modified 2D SSH model was discussed in this paper, we think that a possible extension in 3D SSH on a diamond lattice structure, particularly in the 111 direction, can be useful in explaining our observed results [45].

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
