# Peer review of "Microstructure and Anisotropic Order Parameter of Boron-Doped Nanocrystalline Diamond Films"

_crystals, doi:10.3390/cryst12081031_

Round 1

Reviewer 1 Report

The authors presents microstructure and anisotropic order parameter of boron-doped nanocrystalline diamond films. The research idea is well conceived and the manuscript is well written. The results look good and methods are adequately described. It is very interesting that the author found long-range triplet Josephson current in granular media of the presented nanostructures, which is demonstrated by changing magnetic fields, bias current, temperature and magnetic field angle. However, there are some points that need attention for publication. Hence, I would recommend the authors to revise the manuscript for following comments:

1- Clearly explain the novelty/importance of study in the abstract or introduction sections?

2- Provide the CAS numbers of all the chemicals and compounds used in the experimental section.

3- For readers' understanding, provide a comparative analysis of all the conventional methods of nanocrystalline diamond fabrication and why did the author chose CVD amongst all the methods. Also, what are the implications of different methods of growth provided the boron doping into the diamond structure?

4- Omit all the details regarding Figure 1 in the materials and methods section. The details should only be explained in the results.

5- Provide a schematic image of the device explaining step by step film layers deposited on the substrate along with the device dimensions?

6- It would be great if the author study the structural characteristics of the film via X-ray-diffraction and compare the results with the HRTEM.

7-  I understand that the conclusion is not compulsory for this journal. However, I would strongly recommend the author to provide the limitations to this study and the future directions in the conclusion section for readers' understanding?

Author Response

SBhattacharyya crystals-1778528

The authors presents microstructure and anisotropic order parameter of boron-doped nanocrystalline diamond films. The research idea is well conceived and the manuscript is well written. The results look good and methods are adequately described. It is very interesting that the author found long-range triplet Josephson current in granular media of the presented nanostructures, which is demonstrated by changing magnetic fields, bias current, temperature and magnetic field angle. However, there are some points that need attention for publication. Hence, I would recommend the authors to revise the manuscript for following comments:

Reply: Dear reviewer, thank you very much for your comments. Please find our replies below.

1- Clearly explain the novelty/importance of study in the abstract or introduction sections?

Reply: The microstructure and electrical properties of BNCD films are significantly different from conventional granular superconductors. Such interesting superconducting behavior related to strong spin-orbit coupling in BNCD films has not been found in any other carbon systems such as graphene (since the spin-orbit coupling is weak). Further, we show how a theoretical model can explain the connection between the transport features and the microstructure of BNCD films which can lead to the understanding of a topological insulator in a carbon system.

See lines 28-32, 50-53, and 75-77

2- Provide the CAS numbers of all the chemicals and compounds used in the experimental section.

Reply: The CAS numbers for H2, CH4, and trimethylborane are added on page 2.

See lines 93-95

3- For readers' understanding, provide a comparative analysis of all the conventional methods of nanocrystalline diamond fabrication and why did the author chose CVD amongst all the methods. Also, what are the implications of different methods of growth provided the boron doping into the diamond structure?

Reply: Amorphous diamond-like films (instead of crystalline films) can be deposited by the ion-beam technique. Nanocrystalline diamond (powder instead of films) can be produced by a detonation technique. Single-crystal diamond can be produced by high-temperature high-pressure technique however, doping diamond with light elements, particularly with boron is not easy. This technique is very expensive as well.

Synthesis of high crystalline diamond films requires high energy (temperature) which can be provided by a hot filament CVD technique however, it can introduce contamination in the films from the metal electrodes. Using a plasma CVD technique high-density plasma and very high temperature of electrons can be produced which can produce diamonds of high crystalline order. Most importantly, gas phase doping of diamond can be made in a microwave CVD system with chemicals such as TMB which introduces boron in diamond films. Microwave CVD is a clean technique that does not introduce any metal impurities to diamonds.

See lines 93-95

We have added extra information on the deposition technique in Appendix B.

4- Omit all the details regarding Figure 1 in the materials and methods section. The details should only be explained in the results.

Reply: We agree with the referee on this point. We have moved the text from section 2 to section 3. Thanks for the suggestion.

See lines 122-125

5- Provide a schematic image of the device explaining step by step film layers deposited on the substrate along with the device dimensions?

Reply: Microwave CVD is a standard technique for the deposition of diamond films. Here, we deposit a layer of (polycrystalline) nanocrystalline diamond (film) which is deposited on a 5 mm x 5 mm fused quartz substrate. Columnar growth of diamond nanocrystals is shown in Fig. 1(c). We have added a diagram to Figure 3 and explained the configuration of the device for the transport measurements.

In section 2, a detailed description of the device and the measurements are provided.

See lines 93-95 and 102-107

6- It would be great if the author study the structural characteristics of the film via X-ray-diffraction and compare the results with the HRTEM.

Reply:

The wavelength of electrons (at high applied voltages) is nearly 100 hundred times shorter than x-rays which is ideal for imaging local structures, as shown here. X-ray diffraction does not provide the very local structure of a nanocrystalline film since the crystals are randomly oriented. In this paper, we wanted to look at the atomically thin layer of the grain boundary regions which requires high magnifications. In this work, we have achieved 25 million times of magnification to image the grain boundaries directly which is not possible by x-ray diffraction. Hence, the reflection of the Cooper pairs from the grain boundaries can be matched with the transport properties.

7-  I understand that the conclusion is not compulsory for this journal. However, I would strongly recommend the author to provide the limitations to this study and the future directions in the conclusion section for readers' understanding?

Reply: This is a very useful suggestion. We have added the following to the end of 3.5.2 before the discussion:

Grain boundaries being atomically thick cannot be imaged properly even ultra-high resolution transmission electron microscopy has been employed. It is difficult to find the exact nature of the bonding arrangements at the interface which can be correlated to the topological features of transport. Nevertheless, this is the first report, showing the application of superconducting diamond films in quantum devices where the junction properties can be controlled by the application of a field. We believe that the potential application of carbon-based topological qubits. We think that in carbon systems only diamond has these potential applications.

See lines 404-410, 414-421, and 423-426.

Thanks again for your valuable comments.

Reviewer 2 Report

Substantially valuable work. Language and organization need serious improvement. Drawings careless and partially illegible. Here are just a few remarks

Line 33

Explain what a granular superconductor is

Line 33

doping with light elements in a gas phase

Line 42

Since the microstructure of the diamond can be complicated…. what a complication is it???

Line 60

explain difference between p-junction and p phase  

Line 101

triangular phases ? what the author means

line 109

twins strongly affect the properties of the grain boundaries….- what properties?

Line 111

boron has a preferred growth direction in the diamond

Fig. 1 (b) does not show this Fig1f. it's hard to see, and if the lines shown here suggest rather pentagons, not hexagons

Fig. 2 (b), (d), and (f)]. These dislocations are not shown, can indicate with appropriate markers?

Line 154

All of these studies indicate that the electronic transport of granular diamond systems …. what is granular transport in diamond

Line161

grain boundary junction - what is this?

Fig.4b explain what the arrows mean

Line 311

Rewrite the sentence „theoretical model explains all these microstructural features and electrical properties has not been available

line 374

Pr –probability, Fig.6c- Pr-magnetoresistance???

Author Response

SBhattacharyya crystals-1778528

Substantially valuable work. Language and organization need serious improvement. Drawings careless and partially illegible.

Here are just a few remarks

Reply: Dear reviewer, thank you very much for your comments. Based on your comments we have improved the manuscript significantly (see changes highlighted in the revised version). In particular, figures 1(c), 2(e), and 5(c) &(d) are thoroughly improved.

Please find our replies below.

Line 33

Explain what a granular superconductor is

Reply: We have added a few lines at the beginning of the introduction to explain the granular superconductor.

In detail, we can say that the granularity of the superconducting system can be vital in controlling many of the transport properties, such as critical temperature, localization radius at the boron site, normal state resistance, the coherence length, and Hall mobility as well as the HC2-T phase boundary. These are all grain-size dependent and the grain size can be controlled through synthesis parameters. Hence, it offers a great deal of control over the physical properties of the system. These features have also been observed in granular superconducting systems where superconducting islands are separated by fine-grain boundaries acting as Josephson junctions.

The formation of this so-called anomalous bosonic insulator was ascribed to be due to confinement effects thanks to the granular nature of the system.  In this system, the insulating state is a result of a Coulomb blockade or repulsion between superconducting regions analogous to granular material. 

Our investigations into the diamond microstructure revealed that the nanocrystalline diamond film morphology is fundamentally different from other granular superconducting systems.  This is because, in the case of a superconducting diamond, the boron interfacial material forms a sub-system that alternates with the diamond system in a somewhat ordered fashion similar to a superlattice, in conventional granular superconducting material there is usually no additional subsystem separating the individual superconducting grains. 

See lines 28-32 and 50-53.

Line 33

doping with light elements in a gas phase

Reply: the films are doped with boron which is lighter than carbon.

Line 42

Since the microstructure of the diamond can be complicated…. what a complication is it???

Reply: It is explained as the presence of non-sp3 carbon or a mixture of sp2/sp3 bonds. The line is modified in the introduction.

See lines 47-49.

Line 60

explain difference between p-junction and p phase  

Reply: it should be a p-wave instead of a p-phase. Corrected!

Line 101

triangular phases? what the author means

Reply: It should be a triangular shape (corrected) of the microstructures. A triangle introduces a pi-phase to the system.

line 109

twins strongly affect the properties of the grain boundaries….- what properties?

Reply: Transport properties; particularly a phase transition as shown through the peak in the resistance vs. temperature.

Line 111

boron has a preferred growth direction in the diamond

Fig. 1 (b) does not show this Fig1f. it's hard to see, and if the lines shown here suggest rather pentagons, not hexagons

Reply: Indeed, boron atoms cannot be detected from the TEM. However, grain boundaries have a preferred growth direction as seen in Fig. 1 (b) which can accommodate boron atoms and clusters. The figure has been modified (lines are now removed).

Fig. 2 (b), (d), and (f)]. These dislocations are not shown, can indicate with appropriate markers?

Reply: The figures 2(b) and (d) show the grain boundaries and Fig. 2(f) shows dislocations.

Line 154

All of these studies indicate that the electronic transport of granular diamond systems …. what is granular transport in diamond

Reply: It is the transport in the granular superconductors which shows tunnel transport through the junction. We have modified the sentence.

Line161

grain boundary junction - what is this?

Reply: The junction is created at the interface of two diamond grains i.e. the grain boundary region. However, we have deleted the word ‘junction’ from the revised version.

Fig.4b explain what the arrows mean

Reply: It is explained in the caption of Fig. 4(b). The small arrows show the spin structures.

See lines 257-259

Line 311

Rewrite the sentence „theoretical model explains all these microstructural features and electrical properties has not been available

Reply: The line is changed to “At present, a theoretical model explaining all these microstructural features and electrical properties is not available.”

line 374

Pr –probability, Fig.6c- Pr-magnetoresistance???

Reply: This is correct. The return probability of electrons is equivalent to the resistance which is plotted with the detuning time (or the magnetic field) as explained in Ref. 46 and 47.

Reviewer 3 Report

In the following manuscript, the author is investigating the Microstructure and Anisotropic order parameter of Boron-doped Nanocrystalline Diamond Films. The author has used ultra-high-resolution transmission electron microscopy to reveal the internal symmetry of the grain boundary structure of heavily boron-doped Nanocrystalline diamond films in this study. The most important result is the interface states of the diamond films can be explained by the well-known Shockley model describing the layers connected by vortex-like structures hence forming a topological insulator. Therefore, I recommend the paper for publication in crystals with minor comments below.

Comment 1.

Are all the measurements carried out on single layer Boron-doped Nanocrystalline Diamond Film? In the case of multilayer, how many layers were deposited?

Comment 2.

Figure 5b the oscillatory magnetoresistance, at what temperature data has been taken? 

Author Response

SBhattacharyya crystals-1778528

Comments and Suggestions for Authors

In the following manuscript, the author is investigating the Microstructure and Anisotropic order parameter of Boron-doped Nanocrystalline Diamond Films. The author has used ultra-high-resolution transmission electron microscopy to reveal the internal symmetry of the grain boundary structure of heavily boron-doped Nanocrystalline diamond films in this study. The most important result is the interface states of the diamond films can be explained by the well-known Shockley model describing the layers connected by vortex-like structures hence forming a topological insulator. Therefore, I recommend the paper for publication in crystals with minor comments below.

Reply: Dear reviewer, thank you very much for your comments.

Comment 1.

Are all the measurements carried out on single layer Boron-doped Nanocrystalline Diamond Film? In the case of multilayer, how many layers were deposited?

Reply: The poly(nano)crystalline diamond films are 100 nm thick and have a columnar structure. The microscopy was made in the cross-sectional view which identifies the grain boundaries of the diamond films. We have added more information about the films in section 2 and a new figure (see Fig. 1(c)).

Comment 2.

Figure 5b the oscillatory magnetoresistance, at what temperature data has been taken? 

Reply: It is measured at 2.2 K which is close to the superconducting transition temperature of the films.

Round 2

Reviewer 1 Report

The authors have provided a detailed response to the revision comments. The revised manuscript is improved a lot and now looks ready for publication. Hence, I would like to accept the manuscript publication in the present form.

Author Response

Dear Referee, 

Thank you very much for your review. We have made changes in the manuscripts as you suggested.

Reviewer 2 Report

Thank you for the changes and explanations made

Author Response

Dear Referee, 

Thank you very much for your review. We have made changes in the manuscripts as you suggested. We have checked the English language and style.